# Response to Transforaminal Epidural Block as a Useful Predictive Factor of Postherpetic Neuralgia

**DOI:** 10.3390/jcm8030323

**Published:** 2019-03-07

**Authors:** JungHyun Park, Su Jin Baek, So Hye Baek, Eung Don Kim

**Affiliations:** 1Department of Anesthesiology and Pain Medicine, Incheon St. Mary’s Hospital, College of Medicine, The Catholic University of Korea, Seoul 21431, Korea; happyjj@catholic.ac.kr; 2Department of Anesthesiology and Pain Medicine, Daejeon St. Mary’s Hospital, College of Medicine, The Catholic University of Korea, Seoul 34943, Korea; iamsuza@naver.com (S.J.B.); shye423@gmail.com (S.H.B.)

**Keywords:** herpes zoster, postherpetic neuralgia, predictive factor, epidural block, transforaminal

## Abstract

Despite the high frequency of nerve blocks in the acute phase of herpes zoster, factors associated with intervention, such as response to epidural block, have not been analyzed as predictive factors of postherpetic neuralgia (PHN). To determine the predictive factors of progression to PHN in the presence of interventions, we analyzed the medical records of 145 patients who underwent transforaminal epidural injection (TFEI) in the acute phase of herpes zoster. A total volume of 5 mL (a mixture of 0.5% lidocaine and 5 mg dexamethasone) was injected during TFEI. Corticosteroid was used only for the first TFEI. Clinical data of age, sex, involved dermatome, presence of comorbidity, time from zoster onset to first TFEI, numerical rating scale (NRS) before TFEI, NRS at 1 week and 1, 3, and 6 months after the first TFEI, and number of TFEI were collected and analyzed. Through multivariate logistic regression analysis, pain improvement less than 50% at 1 week after the first TFEI was a strong predictive factor of progression of PHN at all time points. Response to TFEI appears to be a stronger predictive factor of progression to PHN than patient factors of sex, age, degree of initial pain, and presence of co-morbidity.

## 1. Introduction 

Herpes zoster is a disease caused by reactivation of dormant varicella zoster virus (VZV). VZV latent in the dorsal root ganglion (DRG) is replicated and spread along the nerve, causing pain and skin lesion following the corresponding dermatome [1]. The acute stage of herpes zoster is considered as up to one month after skin rash onset. When pain persists beyond the acute phase, neuropathic pain processing, such as central sensitization, occurs and is thought to be one of the major causes of postherpetic neuralgia (PHN) [2,3,4]. 

PHN is the most common complication of herpes zoster and has a reported incidence of 10 to 34%, depending on definition [5,6,7]. PHN is an intractable neuropathic pain condition that can reduce quality of life. The exact discriminative time point of PHN has not been fully agreed yet; however, it is considered to be a PHN when pain persists for more than 3 months after zoster onset [2,4,8]. 

Various clinical features, including age over 50 years, presence of comorbidity, severe pain, and extensive skin lesion, are known predictive factors of PHN [2,4,5,6,7,8,9,10]. However, in actual clinical settings, various nerve blocks are applied for zoster-associated pain (ZAP) control to prevent progression to PHN during the acute phase of herpes zoster. Epidural block is a commonly used intervention method in patients with herpes zoster, and the effect of transforaminal epidural injection (TFEI), which has excellent accessibility to the DRG, has been reported [11]. 

Although nerve blocks are often applied during the acute phase of herpes zoster, factors associated with intervention, such as response to epidural block, have not been analyzed as predictive factors of PHN.

The aim of this study was to determine the predictive factors of progression to PHN in the presence of interventional factors. For this purpose, we analyzed the medical records of patients who underwent TFEI in the acute phase of herpes zoster.

## 2. Method

### 2.1. Participants 

Permission to conduct this study was obtained from the Institutional Ethics Committee of Daejeon St. Mary’s Hospital, Republic of Korea (DC19OESI0006). The medical records of patients who underwent TFEI within one month of zoster onset to control ZAP at Daejeon St. Mary’s Hospital Pain Center from April 2014 and June 2018 were collected. 

TFEI was performed on zoster patients with zoster involving the lumbosacral to thoracic dermatome and who complained of moderate to severe ZAP. Cases of zoster involving the cervical dermatome, facial zoster, infection at the site of injection, coagulopathy, and pregnancy were not indicated for TFEI.

All patients were prescribed analgesics, such as aceclofenac and tramadol/acetaminophen combination tablets. When neuropathic features of electric shock-like pain or allodynia were observed, pregabalin and nortriptyline were used as an anticonvulsant and tricyclic antidepressant, respectively. 

All participants received appropriate antiviral treatment. The medical records of patients who did not receive appropriate antiviral agents within 72 hours after skin rash were excluded from the study. Medical records from patients who had received the first TFEI beyond one month of zoster onset were also excluded from this analysis. 

### 2.2. Procedure

Patients were placed in a prone position on a fluoroscopic-compatible table. The needle entry site was prepared and draped in a sterile fashion. A fluoroscope was placed obliquely toward the ipsilateral side. After local infiltration, a 22-G Quincke needle was introduced inferior to the pars interarticularis under fluoroscopic guidance at the level of pathology.

The needle tip position was adjusted to inferior to the pedicle in the anteroposterior view. To minimize the possibility of vascular injury, the needle tip was positioned in the lower posterior portion of the intervertebral foramen in the lateral view of the fluoroscopic image (Figure 1).

A total of 2 to 3 mL of contrast agent (Iobrix inj. 300, Taejoon Pharm., Seoul, Korea) was used to confirm needle tip position. Then, 5 mL of a mixture of 0.5% lidocaine and 5 mg dexamethasone was injected. Corticosteroid was added to the injectate only for the first administration of TFEI. All TFEI procedures were conducted by a physician experienced in pain management (E.D.K.).

### 2.3. Data Collection

The following data were collected from medical records and analyzed: age, sex, involved dermatome, presence of comorbidity, time from zoster onset to first TFEI, numerical rating scale (NRS) before TFEI, NRS at 1 week and 1, 3, and 6 months after TFEI, and number of TFEI procedures performed during the review period.

### 2.4. Outcome Measure

PHN was defined as the presence of pain with an NRS score of 1 or higher. The examination time points were set at 1 month, 3 months, and 6 months after the first TFEI. The presence of PHN and variables of interest at these time points were evaluated. 

Pain intensity prior to TFEI (baseline pain) was classified as moderate pain between NRS 3 and 6 and severe pain from NRS 7 to 10. Characteristics of baseline pain were classified according to the presence of neuropathic features.

Age was classified as 50 years or older and less than 50 years. Comorbidity was categorized as cardiovascular, diabetes, two or more diseases combined, and no underlying disease.

The time of TFEI was classified as having received the first TFEI within 15 days after zoster onset or between 16 and 30 days after zoster onset. 

We asked patients to revisit our pain center one week after TFEI for ZAP assessment using the NRS. The response to TFEI was divided into cases of pain improvement of 50% or more (remaining pain ≤ 50% of baseline NRS) or pain improvement less than 50% (remaining pain > 50% of baseline NRS) when evaluated at 1 week after the first TFEI. 

If the intensity of the worst pain over the last 24 h was greater than NRS 3, we recommended repetition of TFEI. The number of TFEI performances was categorized as once or more than once. 

### 2.5. Statistical Analysis

Demographic data of the patients were analyzed using Student’s *t*-test for continuous variables and Chi-square or Fisher’s exact test for categorical variables. A *p*-value less than 0.05 was considered statistically significant.

To determine independent risk factors associated with development of PHN among the clinical data and demographic characteristics of the participants, binary logistic regression techniques were used.

Variables were screened by examining for multicollinearity (variables were excluded if correlation coefficient *r* > 0.7) and were further selected with a forward stepwise procedure (entry criteria *p* = 0.05, removal criteria *p* = 0.10). All data were analyzed using SPSS version 18.0 (SPSS Inc, Chicago, IL, USA). All parametric data are presented as the mean ± standard deviation (SD) and nonparametric data as number and proportion. Odds ratio (OR) with 95% confidence interval (CI) was also calculated as needed.

## 3. Results 

We collected 158 medical records of patients who underwent TFEI in the acute phase of herpes zoster. Of these, 13 patients were excluded from the analysis due to insufficient medical records. Finally, medical records of 145 patients were used for the analysis. The demographic data of the patients are described in Table 1. 

The mean age of the patients was 61.95 ± 13.47 years. There were many more women (*n* = 104) than men (*n* = 41). The mean baseline NRS prior to the first TFEI was 6.41 ± 1.27, and the mean time of receiving the first TFEI after zoster onset was 10.47 ± 6.84 days.

Approximately 79% of patients developed herpes zoster in the thoracic dermatome. More than half of patients (59.3%) had no underlying disease. Neuropathic pain features were observed in almost all patients (97.9%). 

### 3.1. Factors Associated with PHN in Univariate Analysis

The PHN ratio was 36.5% at 1 month, 9.6% at 3 months, and 6.8% at 6 months after the procedure. At all of the time points, the proportion of cases with pain improvement less than 50% at 1 week after the first TFEI was significantly higher in PHN patients than non-PHN patients (1 month: 25/53, 47.2% vs. 12/92, 13.0%; *p* < 0.0001, 3 months: 9/14, 64.3% vs. 28/131, 21.4%; *p* = 0.008, 6 months: 6/10, 60.0% vs. 31/135, 22.9%; *p* = 0.010).

At 1 month, the proportion of patients over 50 years of age (50/53. 94.3% vs. 71/92, 77.2%; *p* = 0.022) and the ratio of severe baseline pain intensity (34/53, 64.2% vs. 37/92, 40.2%; *p* = 0.014) were significantly higher in PHN patients than non-PHN patients. At 6 months, the proportion of patients who received the first TFEI beyond 15 days of zoster onset was significantly higher in the PHN patients than non-PHN patients (5/10, 50.0% vs. 28/135. 20.7%; *p* = 0.027) (Table 2).

### 3.2. Factors Associated with PHN in Multivariate Logistic Regression

The results of multivariate logistic regression are described in Table 3. Age over 50 years (OR = 0.14; 95% CI, 0.03–0.63; *p* = 0.010) and severe baseline pain (OR = 3.38; 95% CI, 1.44–7.87; *p* = 0.005) were significant predictive factors of PHN at 1 month. Performance of the first TFEI beyond 15 days after zoster onset was a significant predictive factor of PHN at 6 months (OR = 5.82; 95% CI, 1.28–26.45; *p* = 0.023). At all of the time points, pain improvement less than 50% at 1 week after the first TFEI was a strong predictive factor of progression of PHN (1 month: OR = 8.27; 95% CI, 3.07–22.27; *p* < 0.0001, 3 months: OR = 6.96; 95% CI, 2.12–22.85; *p* = 0.001, 6 months: OR = 7.14; 95% CI, 1.63–31.28; *p* = 0.009).

## 4. Discussion

In the present study, less than 50% improvement at 1 week after the first TFEI was a strong predictive factor of progression of PHN at all time points. The time to first TFEI after zoster was an additional significant PHN predictive factor at 6 months. 

Previous studies have reported age over 50 years, female gender, comorbidity, extensive skin lesion, and severe pain in the acute phase as predictive factors of PHN progression [2,4,5,6,7,8,9,10]. However, these studies focused only on patient factors and did not consider the performance of nerve blocks in the acute phase; therefore, the limited predictive factors mentioned above might be the result of not considering the effect of the intervention. In an actual clinical setting, various interventions are commonly attempted to control ZAP; therefore, it would be meaningful to include the responses to these interventions in the analysis to identify predictive factors of PHN. 

Epidural block has been commonly used to control ZAP in the acute phase and prevent progression to PHN; however, there is still controversy about its efficacy to prevent PHN [12,13,14,15]. In previous epidural block studies, the interlaminar approach was used. As the interlaminar approach is performed in a blind manner, there is a possibility that a sufficient amount of administered drug may not reach the affected DRG [13,16]. Moreover, even under fluoroscopic guidance, the interlaminar approach at the thoracic level may be technically challenging due to the steep angle of the spinous process of the thoracic vertebra and the relatively small size of the interlaminar foramen [11].

Considering that VZV is reactivated in the DRG, this would be an appropriate target lesion for intervention at the time of acute zoster [1,17,18]. The transforaminal approach has higher accessibility to the DRG than the interlaminar approach, and its positive effect on ZAP control has been reported [11].

In this study, patient factors such as baseline pain intensity and age of 50 years or older were significant predictors of PHN in only the relatively early period, such as 1 month after the procedure. However, such patient factors were no longer significant when predicting PHN at 3 months after the procedure. In contrast, at all of the time points, regardless of age, comorbidity, severity of baseline pain, and initial neuropathic features, the response to TFEI was a strong factor to predict progression to PHN. This suggests that suppression of the nociceptive signal in neuronal tissues, such as the DRG, is very important in inhibiting progression to PHN. Perhaps the degree of blockage of nociceptive signal might be more influential on progression to PHN than patient factors such as age, sex, intensity of initial pain, and presence of comorbidity. 

One of the limitations of present study was that it was unable to strictly control factors such as dose of analgesics or anticonvulsant due to its retrospective nature. Another limitation is that the analysis was based on a relatively small sample size collected from a single hospital. However, this study was the first to investigate the predictive factors of progression to PHN, including the presence of intervention, such as TFEI, and more likely to be performed in close proximity to the actual clinical situation.

## 5. Conclusions 

In conclusion, the response to TFEI in the acute phase of herpes zoster appears to be a strong predictive factor of progression to PHN. We believe that if no meaningful effect is observed during follow-up of patients after TFEI, another type of intervention, such as continuous epidural block or pulsed radiofrequency, should be considered so as not to delay suppression of nociceptive signaling to the central nervous system. Based on our findings, further prospective studies are needed to develop a more appropriate treatment protocol for ZAP control and prevention of PHN.

## Figures and Tables

**Figure 1 jcm-08-00323-f001:**
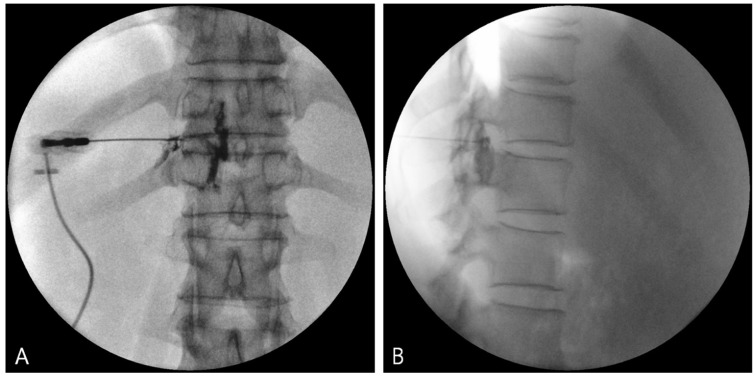
Fluoroscopic images of TFEI. (**A**) Anteroposterior (AP) view. (**B**) Lateral view. TFEI: Transforaminal epidural injection.

**Table 1 jcm-08-00323-t001:** Demographic data of participants.

Variables	Results
Age, years, mean ± SD	61.95 ± 13.47
Sex, *n* (male/female)	41/104
Direction, *n* (Right/Left)	73/72
Baseline NRS	6.41 ± 1.27
Timing of 1st TFEI, days, mean ± SD	10.47 ±6.84
Presence of neuropathic feature, *n* (%)	142 (97.9)
Dermatome, *n* (%)	
Thoracic	114 (78.6)
Lumbosacral	31 (21.4)
Comorbidity, *n* (%)	
Cardiovascular	31 (21.4)
Diabetes	13 (9.0)
Combined	15 (10.3)
None	86 (59.3)

SD: standard deviation; NRS: numerical rating scale; TFEI: Transforaminal epidural injection.

**Table 2 jcm-08-00323-t002:** Univariate analysis of PHN vs. non-PHN at 1 month, 3 months, and 6 months after the procedure.

	1 Month				3 Months				6 Months			
Variables	No PHN (*n* = 92)	PHN (*n* = 53)	*p*-Value	Odds Ratio (95% CI)	No PHN (*n* = 131)	PHN (*n* = 14)	*p*-Value	Odds Ratio (95% CI)	No PHN (*n* = 135)	PHN (*n* = 10)	*p*-Value	Odds Ratio (95% CI)
Sex (male/female)	29/63	12/41	0.317	1.67 (0.61–4.57)	35/96	6/8	0.491	0.61 (0.15–2.50)	36/99	5/5	0.13	0.25 (0.038–1.53)
Direction (Right/Left)	46/46	27/26	0.720	1.178 (0.482–2.87)	66/65	7/7	0.78	1.23 (0.29–5.22)	68/67	5/5	0.98	1.02 (0.16–6.43)
Age												
≥50	71	50	0.022 *	0.134 (0.024–0.745)	107	14	0.99	-	111	10	0.99	-
<50	21	3			24	0			24	0		
Comorbidity												
Cardiovascular	18	13	0.383	1.61 (0.56–4.64)	27	4	0.45	2.00 (0.34–11.85)	29	2	0.99	-
Diabetes	5	8	0.069	3.56 (0.91–13.94)	13	0	0.99	-	13	0	-	-
Combined	11	4	0.764	0.81 (0.43–2.90)	12	3	0.21	3.37 (0.49–22.88)	12	3	0.99	-
None	58	28	0.83	1.11 (0.43–2.90)	79	7	0.77	1.29 (0.23–7.14)	81	5	0.99	-
Baseline pain												
Moderate (NRS 3–6)	55	19	0.014 *	3.06 (1.25–7.52)	87	7	0.66	0.72 (0.16–3.13)	70	4	0.69	1.44 (0.24–8.61)
Severe (NRS ≥ 7)	37	34			64	7			65	6		
Timing of the first TFEI												
Within 15 days	75	37	0.132	2.08 (0.80–5.39)	103	9	0.15	3.19 (0.67–15.21)	107	5	0.027 *	7.46 (0.25–44.49)
16th–30th days	17	16			28	5			28	5		
Dermatome												
Thoracic	72	42	0.755	1.18 (0.41–3.47)	104	10	0.18	3.04 (0.61–15.34)	108	6	0.063	5.58 (0.91–34.17)
Lumbosacral	70	11			27	4			27	4		
Remaining pain at 1 week after the first TFEI (Response to TFEI)												
≤50% of baseline NRS (pain improvement of 50% or more)	80	28	<0.0001 ^#^	9.13 (3.19–26.10)	103	5	0.008 *	7.39 (1.67–32.79)	104	4	0.010 *	7.67 (0.85–68.83)
>50% of baseline NRS (pain improvement less than 50%)	12	25			28	9			31	6		
Presence of neuropathic feature												
Exist	89	53	0.626	0.85 (0.43–1.66)	107	14	0.99	-	132	10	0.99	-
Do not exist	3	0			24	0			3	0		
Number of TFEI												
Once	63	22	0.117	1.994 (0.84–4.72)	80	5	0.37	1.98 (0.45–8.86)	80	5	0.37	0.34 (0.03–3.72)
More than twice	29	30			50	9			54	5		

PHN: Postherpetic neuralgia; NRS: Numerical rating scale; *: *p* < 0.05, ^#^: *p* < 0.0001.

**Table 3 jcm-08-00323-t003:** Multivariate logistic regression analysis of independent risk factors associated with the development of PHN.

Variables	Odd Ratio	95% CI	*p*-Value
1 month			
Age ≥ 50	0.14	0.03–0.63	0.010 *
Baseline severe pain intensity (NRS ≥ 7)	3.38	1.44–7.87	0.005 *
Remaining pain > 50% of baseline NRS at 1 week after the first TFEI	8.27	3.07–22.27	<0.0001 ^#^
3 months			
Remaining pain > 50% of baseline NRS at 1 week after the first TFEI	6.96	2.12–22.85	0.001 *
6 months			
Timing of the first TFEI (16^th^–30th days after zoster onset)	5.82	1.28–26.45	0.023 *
Remaining pain > 50% of baseline NRS at 1 week after the first TFEI	7.14	1.63–31.28	0.009 *

NRS: Numerical rating scale. *: *p* < 0.05, ^#^: *p* < 0.0001

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
