# Peer review of "Response to Transforaminal Epidural Block as a Useful Predictive Factor of Postherpetic Neuralgia"

_jcm, 2019, doi:10.3390/jcm8030323_

Reviewer 1 Report

The study by Park and colleagues aimed to determine the predictive factors of progression of herpes zoster (HZ) disease to postherpetic neuralgia (PHN) in the presence of interventional factors, such as response to epidural block. To do this, the authors retrospectively analyzed the medical records of 145 patients who underwent transforaminal epidural injection (TFEI) in the acute phase of HZ. Clinical data of age, sex, involved dermatome, presence of underlying disease, time from HZ onset to first TFEI, numerical rating scale (NRS) before TFEI, NRS at 1 week and 1, 3, and 6 months after the first TFEI, and number of TFEI procedures were collected and analyzed. It was found that less than 50% pain improvement at 1 week after the first TFEI was the strongest predictive factor of progression of HZ to PHN at all time points. The time to first TF after HZ was also an additional significant predictive factor of progression of HZ to PHN at 6 months. Patient factors of sex, age, degree of initial pain, and presence of co-morbidity were less significant predictive factors of progression of HZ to PHN. The results suggests that response to TFEI in the acute phase of HZ is the strongest predictive factor of progression to PHN, thereby suggesting the importance of further blocking the nociceptive signal to the central nervous system by the use of other methods and/or interventions (e.g. pulsed radio-frequency, continuous epidural block etc).

This is an excellent manuscript. The findings of the study are novel and interesting, and could prove to be clinically useful. Limitations include small sample size, lack of blinding, and an arguably low criterion of NRS (1) for PHN. Future larger (prospective) studies are needed to develop more appropriate treatment protocols for preventing progression of HZ to PHN. The findings of this paper would definitely be of interest to the readership of the journal.

Author Response

Response) We sincerely appreciate your kind comments.

Reviewer 2 Report

Patients suffering from herpes zoster (HZ) secondary to resurgence of VZV in a (single) dorsal root ganglion (DRG) are at risk of going on to develop the chronic, painful neuropathic pain condition postherpetic neuralgia (PHN).  A number of patient-related risk factors for  this progression are known. The authors propose a new one.  They asked whether response to epidural deposition of lidocaine and dexamethasone in the segment of the infected DRG at the HZ stage correlates with the likelihood of developing PHN. The research was based on a retrospective review and multivariate statistical analysis of medical files of 145 HZ patients who received injections. The authors concluded that patients injected soon after HZ onset and who obtained good pain relief were considerably less likely to develop PHN than those who didn’t. Patient injected > 15d after HZ onset and obtained good pain relief also showed less PHN, but the effect was considerably less.

The authors acknowledge some of the intrinsic weaknesses of small retrospective analysis of this sort, although not the lack of blinding and the absence of a true control group. Nonetheless it is moderately convincing that a good response to early block has some prognostic value.  This is a novel finding to the best of my knowledge. However, its clinical usefulness strikes me as modest in the sense that a patient with a good early response still runs a risk of developing PHN. The prognosis obtained is not certain enough to justify abandoning further treatment except perhaps under very special circumstances. Likewise, much stronger evidence would be required to no justify shifting patients without a good early response to unproven modalities such as pulsed radiofrequency.

The result does provide a novel bit of data towards the better understanding of the mechanism of HZ/PHN pain. In this regard I am very disappointed that the authors failed to report the immediate effect (a few hours) of injection, while the lidocaine was still acting. Their first readout was at one week. It is not certain whether the nerve impulses that cause background (spontaneous) pain in HZ (and PHN) originate in the periphery, the infected ganglion, the CNS or elsewhere. It would be very good to know whether 0.5% lidocaine deposited on the surface of the infected DRG stops HZ pain transiently. This concentration is probably too dilute to block propagation of impulses from the periphery as they pass through the foramen, while it is probably sufficient to stop the ectopic generation of impulses within the DRG.  If the medical records include information on the acute effect of the lidocaine on HZ pain, adding this information to the MS now would certainly increase its value added to readers. Acute post-injection pain relief may also have prognostic value, maybe even more than relief at 1 week. A discussion of pain mechansisms in HZ and PHN is available in the latest edition (2016?)  of the Watson et als. book on HZ and PHN.

In their Discussion the authors go from correlation (prognosis) to a statement about the mechanism underlying their observations (lines 251-260). Specifically, they suggest that their intervention, which they presume suppressed nociceptive signals, was a causative factor in reducing the likelihood of transition to PHN. This is not unreasonable, but the supporting evidence base from this study is very weak. It is more likely that the responders had less disease (or a more effective immune system) from the beginning, and that 1 week response to injection just served as a screen (biomarker) rather than actually being therapeutic. The large RCTs cited on the efficacy of corticosteroids on development of PHN are not very optimistic on this point. These patients probably had a better prognosis from the beginning. A qualifying statement about this should be added to the Discussion.

Minor issues:

1) In the Abstract please mention what drugs you inject epidurally.

2)  “Comorbidity” may be a better term than ”underlying disease”, which sounds like a disease that causes the PHN pain. Make clear that only two comorbidities among many potential ones were scored (assuming I’ve understood this right) and maybe replace “combined” with “both”.   

3) NRS=1 is a very low criterion for PHN. Would the result have been different if a more debilitating threshold had been chosen?

4) The English is good and fully comprehensible, but there are a few odd phrases that a native English speaker might pick up.

Author Response

Patients suffering from herpes zoster (HZ) secondary to resurgence of VZV in a (single) dorsal root ganglion (DRG) are at risk of going on to develop the chronic, painful neuropathic pain condition postherpetic neuralgia (PHN).  A number of patient-related risk factors for  this progression are known. The authors propose a new one.  They asked whether response to epidural deposition of lidocaine and dexamethasone in the segment of the infected DRG at the HZ stage correlates with the likelihood of developing PHN. The research was based on a retrospective review and multivariate statistical analysis of medical files of 145 HZ patients who received injections. The authors concluded that patients injected soon after HZ onset and who obtained good pain relief were considerably less likely to develop PHN than those who didn’t. Patient injected > 15d after HZ onset and obtained good pain relief also showed less PHN, but the effect was considerably less.

The authors acknowledge some of the intrinsic weaknesses of small retrospective analysis of this sort, although not the lack of blinding and the absence of a true control group. Nonetheless it is moderately convincing that a good response to early block has some prognostic value.  This is a novel finding to the best of my knowledge. However, its clinical usefulness strikes me as modest in the sense that a patient with a good early response still runs a risk of developing PHN. The prognosis obtained is not certain enough to justify abandoning further treatment except perhaps under very special circumstances.

Likewise, much stronger evidence would be required to no justify shifting patients without a good early response to unproven modalities such as pulsed radiofrequency.

Response) Thank you for your good point. Our argument is that if there is no significant symptomatic improvement after TFEI ( poor 1 week response), it is important to try other treatment options such as DRG PRF (Of course, we agree with your opinion that methods like PRF have not yet been fully proven) or continuous epidural block to prevent progression to neuropathic pain.

The result does provide a novel bit of data towards the better understanding of the mechanism of HZ/PHN pain. In this regard I am very disappointed that the authors failed to report the immediate effect (a few hours) of injection, while the lidocaine was still acting. Their first readout was at one week. It is not certain whether the nerve impulses that cause background (spontaneous) pain in HZ (and PHN) originate in the periphery, the infected ganglion, the CNS or elsewhere. It would be very good to know whether 0.5% lidocaine deposited on the surface of the infected DRG stops HZ pain transiently. This concentration is probably too dilute to block propagation of impulses from the periphery as they pass through the foramen, while it is probably sufficient to stop the ectopic generation of impulses within the DRG.  

If the medical records include information on the acute effect of the lidocaine on HZ pain, adding this information to the MS now would certainly increase its value added to readers. Acute post-injection pain relief may also have prognostic value, maybe even more than relief at 1 week. A discussion of pain mechansisms in HZ and PHN is available in the latest edition (2016?)  of the Watson et als. book on HZ and PHN.

Response) Thank you for your sharp point. We agree with you regarding concentration of lidocaine. In many articles, the concentration of lidocaine was 0.5% or more. However, according to our clinical experience, low concentration of lidocaine even lower than 0.4% can provide sufficient pain relief. And in randomized trial of epidural injection in New England Journal of Medicine 2014;371:11-21, the concentration of lidocaine was 0.25% to 1%. Therefore, 0.5% lidocaine also can be effective to block nocicepetive impulses. In addition, when epidural block was performed at thoracic level, to minimize the risk of hemodynamic change, 0.5% lidocaine was used.

We fully agree with your opinion that acute post injection pain relief (observation of immediate effect) would have prognostic value. Unfortunately, the description of the immediate post-injection effect in the medical record used in this analysis was incomplete. Most of the patients were followed up at weekly intervals, so 1 week response was defined as one of the outcomes.

In their Discussion the authors go from correlation (prognosis) to a statement about the mechanism underlying their observations (lines 251-260). Specifically, they suggest that their intervention, which they presume suppressed nociceptive signals, was a causative factor in reducing the likelihood of transition to PHN. This is not unreasonable, but the supporting evidence base from this study is very weak. It is more likely that the responders had less disease (or a more effective immune system) from the beginning, and that 1 week response to injection just served as a screen (biomarker) rather than actually being therapeutic.

The large RCTs cited on the efficacy of corticosteroids on development of PHN are not very optimistic on this point. These patients probably had a better prognosis from the beginning. A qualifying statement about this should be added to the Discussion.

Response) Thank you for such a good point. As your opinion, it may be reasonable that patients in good health from the beginning were responding well to TFEI. However, interestingly, multivaraite regression analysis of present study found that comobidity was not a significant predictive factor of progression to PHN.

We also totally agree with your opinion that 1 week response can be a screening role. We also want to establish a more effective treatment protocol based on the results of this study.

Minor issues:

In the Abstract please mention what drugs you inject epidurally.

Response) Thank you for your good point. As your recommendation, we have added the following sentence to the abstract:

“A total volume of 5 ml (a mixture of 0.5% lidocaine and 5 mg dexamethasone) was injected during TFEI. Corticosteroid was used only for the first TFEI.”

2)  “Comorbidity” may be a better term than ”underlying disease”, which sounds like a disease that causes the PHN pain. Make clear that only two comorbidities among many potential ones were scored (assuming I’ve understood this right) and maybe replace “combined” with “both”.   

Response) As your recommendation, the term "underlying disease" is changed to "comorbidity" throughout the manuscript.

Regardless of disease type, conditions with more than two diseases were classified as "combined".

3) NRS=1 is a very low criterion for PHN. Would the result have been different if a more debilitating threshold had been chosen?

Response) Although there are various definitions of PHN, it was defined as an NRS score of 1 or higher in many studies (Acta Anaesthesiol Scand 2000;44:910–8, Lancet. 2006;367:219–24, Anesth Analg. 2009;109:1651–5, Pain Pract. 2015;15:229–35., etc.).

Because we also wanted a rigorous standard, we decided to consider NRS 1 or more as PHN as in previous studies.

4) The English is good and fully comprehensible, but there are a few odd phrases that a native English speaker might pick up. 

Response) This manuscript was reviewed by a commercial English-language editing service (eWorldEditing Inc., Eugene, OR, USA).